# The Tale of *Staphylococcus aureus* Isolated from Mastitis Infections: The Effect of Antimicrobials and Bacterial Relatedness

Angela Perdomo [1] , Maria Salazar [1] , Rasmi Janardhanan [2] and Alexandra Calle [1,*]

1   School of Veterinary Medicine, Texas Tech University, Amarillo, TX 79106, USA;
    angela.perdomo@ttu.edu (A.P.); maria.salazar@ttu.edu (M.S.)
2   Institute for Innovation & Sustainable Food Chain Development, Campus de Arrosadía,
    Universidad Pública de Navarra, 31006 Pamplona, Spain; rasmi.janardhanan@unavarra.es
*   Correspondence: alexandra.calle@ttu.edu

**Abstract:** *Staphylococcus aureus* is a common causative agent of mastitis in dairy cattle, posing a substantial threat to animal health and resulting in significant economic losses. Preventive measures are usually in place to control the spread of the organism between animals and around the dairy environment; however, mastitis outbreaks can still be recurrent. During this investigation, a total of 30 *S. aureus* isolates were obtained from six deceased cows, all diagnosed with chronic mastitis during an outbreak in West Texas. The aim of this study was to evaluate the response of the *S. aureus* isolates causing severe mastitis infections to iodine treatments and their antibiotic susceptibility, planktonic growth, and biofilm formation. Udder skin was inoculated with *S. aureus* and subjected to various iodine concentrations of 0.25%, 0.38%, 0.50%, 0.75%, and 1.00%, with exposure times of 15 s, 10 s, and 60 s. The same concentrations were tested on *S. aureus*'s biofilm formation. The results of the antimicrobial susceptibility test indicate that the exposure time did not influence the treatment. Lower iodine concentrations were compared with 1.00%, as the standard treatment used by the dairy for teat disinfection, and statistical difference ($p < 0.001$) was evident in the 0.00% iodine treatment compared to the other iodine concentrations. Moreover, a significant difference ($p < 0.001$) emerged when comparing the 0.25% and 0.38% iodine concentrations with 1.00%. No difference ($p > 0.161$) was detected between 0.50%, 0.75%, and 1.00%. These results suggest that, under the conditions investigated, iodine can be lowered to around 50% of the currently used dose without negatively impacting microbial control. On the other hand, *S. aureus* strains were susceptible to the tested antibiotics, demonstrating that antimicrobial resistance does not always play a role in the persistent mastitis infections caused by *S. aureus*. Further microbial phenotypic typing conducted on *S. aureus* strains indicated a possible common source of the infections, demonstrating the potential of there being resident *S. aureus* strains at this dairy farm.

**Keywords:** *Staphylococcus aureus*; iodine; mastitis; biofilm; microbial typing





## 1. Introduction

Mastitis is one of the most frequently occurring diseases in dairy cattle worldwide [1,2]. This condition is a result of physical trauma or microbial infection, which leads to an inflammation within the mammary gland. It can permanently damage the parenchymal tissue, leading to decreased milk production and secondary carry-over effects such as poor fertility and a high culling rate [3]. The economic impact of mastitis is significant, with an estimated cost of USD 147–179 per cow per year associated with its treatment, reduced milk production, and the disposal of milk [4,5].

Bovine mastitis control requires focusing on preventive measures to avoid the spread of pathogens in the dairy farm environment and between cows. Dairies consistently work to reduce the incidence of new infections, improve the overall health and well-being of their animals, and increase the quality and quantity of milk production [6–9]. Preventive

measures include practices that involve humans, animals, and the environment, such as hygiene and sanitation practices for workers, maintaining a clean farm environment, and regular veterinary check-ups for cows. A commonly used practice is the routine application of pre- and post-milking teat antimicrobials, such as iodine-based teat products, which is highly recommended to reduce the probability of contamination from the milking parlor environment to the cow. Teat disinfection usually occurs by conducting a teat dip using iodine at 1%. Upon milking, udders are always sealed and protected with this antimicrobial solution to prevent microbial attachment and multiplication. This practice is widely accepted to help prevent intramammary infections (IMIs), since iodine-based teat disinfectants have been demonstrated to be effective in controlling some of the most prevalent mastitis-causing bacteria [5,10–12]. Even though these chemicals are effective in controlling *S. aureus* on mammary tissue and preventing it from transferring to milking equipment, reports suggest that prolonged exposure to germicidal teat dips can enhance the resistance of some bacteria to chemical disinfectants [12,13].

An antibiotic treatment is normally administered to cows when chronic mastitis is observed. For example, during dry cow therapy, cows are not milked to the involution, regeneration, and preparation of the mammary gland for a new lactation. This is a strategy that helps prevent and control harmful bacteria from infiltrating the mammary gland and it is one of the most common options for controlling mastitis infections in dairy cows [5,14,15]. In all cases, while animal health is crucial, the use of antibiotics is always a concern due to the risk of antimicrobial resistance in bacterial pathogens [16].

*Staphylococcus aureus* is one of the most prevalent mastitis-causing bacteria reported in dairy cattle [17,18]. The frequency of subclinical and chronic mastitis caused by *S. aureus* varies around the world, and it is influenced by factors such as the stage of lactation, immune status of the animal, stress, vaccination status, season, location of the herd, and overall management, to name a few [4,19]. Bacteria enter the mammary gland through orifice-chapped or injured teats. It attaches to epithelial cell receptors, producing various virulence factors and the intracellular uptake of small colony variants of *S. aureus*. Once the infection takes place, it becomes difficult to treat, and infected cattle become a transmission source [20,21]. Treating *S. aureus* IMIs is challenging, and recovery rates vary considerably, increasing the odds of environmental cross-contamination or cow-to-cow transmission [22,23]. Despite antibiotic treatment, *S. aureus* mastitis is prone to recurrence, and outbreaks in dairy cattle are somewhat common.

An outbreak of mastitis at a dairy farm in the Texas panhandle led to the unfortunate loss of many cows. With the implementation of multi-hurdle interventions, including cow culling and antimicrobial treatments, veterinarians were able to contain the outbreak. This research was conducted to offer scientific insights into some of the mechanisms used to prevent the spread of isolated bacteria from deceased cows affected by the outbreak. The objectives of the present investigation were (i) to explore the effects of varying iodine concentrations on *S. aureus* isolates obtained from intramammary secretions, (ii) to evaluate the antimicrobial responses of the *S. aureus* isolates, and (iii) to determine the relatedness of *S. aureus* isolates across mastitic milk samples and ascertain the presence of the same bacterial strain, potentially indicating its role as a common source during an outbreak.

## 2. Materials and Methods

### 2.1. Staphylococcus aureus's Isolation

Severely infected milk samples were obtained from six culled cows previously diagnosed with a chronic mastitis infection. The attending veterinarian manually extracted the milk secretions from the infected quarters of the already deceased cows, deposited them in sterile cups, and delivered them to the Food Microbiology Laboratory at Texas Tech University (TTU). The dairy farm providing the samples was a medium-to-large operation, located in west Texas, experiencing a mastitis outbreak. Infected cows were segregated in the hospital pen for treatment or culling. Animal segregation served as a means to prevent the spread of the bacterial infection among the herds.

For the isolation of *S. aureus*, a volume of 10 mL from each mastitis secretion was enriched in 90 mL Tryptic Soy Broth (TSB, Millipore Sigma, Burlington, MA, USA) with 10% NaCl and incubated at 37 °C for 24 h. Upon incubation, each sample was streaked onto Mannitol Salt Agar (MSA, HiMedia Laboratories LLC, Philadelphia, PA, USA) and incubated at 37 °C for 24 h. Five presumptive *S. aureus* colonies were isolated from each MSA plate. Each of the 30 isolated colonies was re-streaked individually on Tryptic Soy Agar (TSA, Millipore Sigma, Burlington, MA, USA) and incubated at 37 °C for 24 h. This process was repeated twice to improve isolation. To preserve the isolates for the duration of this research, each colony was inoculated in Tryptic Soy Broth (TSB, Millipore Sigma, Burlington, MA, USA) and incubated at 37 °C for 18 h in an incubator shaker. After incubation, isolates were cryopreserved at −80 °C for in-house culture collection and further testing.

All *S. aureus* isolates were confirmed using real-time PCR or Matrix-assisted laser desorption/ionization-time of flight (MALDI-TOF) mass spectrometry (MS). In brief, for the PCR confirmation, a single colony was inoculated in 9 mL of TSB and incubated at 37 °C for 24 h. The Hygiena™ BAX® System Real-Time PCR *Staphylococcus aureus* assay was conducted, following the manufacturer's guidelines. This protocol requires a DNA extraction step followed by the PCR reaction using, in all cases, their proprietary kits. In the case of isolates not being identified or confirmed using the previous method, MALDI-TOF was used. For this method, a cryopreserved culture was streaked onto TSA plates and incubated at 37 °C for 24 h. A single colony obtained from TSA was placed onto a plate spot, and 1 μL of α-Cyano-4-hydroxycinnamic acid (CHCA) matrix was added on top of the colony. Then, the spectra were analyzed using the Confidence Axima system from Shimadzu Corporation (Shimadzu Scientific Instruments, Inc, Columbia, MD, USA), cross-referencing its identification in Shimadzu Launchpad software (Ver 2.32 Shimadzu Corporation, Kyoto, Japan) and using the VITEK® MS (Ver 2.9.8.1 Shimadzu Corporation, Kyoto, Japan) SARAMIS v4.13.0 RUO database.

Thus, from the six original infected milk samples, a total of 30 isolates (5 isolates per sample) were recovered. From these 30 isolates, a subset of 18 (3 isolates per each of the six samples) were used in some of the experiments, as described further.

### 2.2. Testing Iodine Treatments on Udder Skin with S. aureus Planktonic Cells

For this portion of the study, three isolates were selected from each of the six original mastitic milk samples ($n = 18$). Udder skin was obtained from a beef slaughter facility and was used to test iodine treatments by inoculating the skin with each *S. aureus* strain. To prepare the udder skin for the experiments, any presence of hair was removed from the top of the skin, as well as excess fat from underneath. The skin was cut into squared portions of about 10 g. To sanitize prior treatment, samples were immersed in boiling water for 15 s, allowed to air dry, and cooled down to ambient temperature. The skin samples were inoculated with 150 μL of an overnight culture of *S. aureus*. The inoculum was evenly spread on the udder samples using a cotton swab and air dried on the laboratory benchtop for 20 min to facilitate bacterial attachment. Each inoculated sample underwent random treatment in terms of a combination of dip time and iodine concentration. The inoculated skin was treated for 15 s, 30 s, and 60 s using commercial teat dip solutions containing 0.25%, 0.38%, 0.50%, 0.75%, and 1.00% iodine, with each dip solution containing 1% titratable iodine (Dairyland Brand from Stearns Packaging Corporation, Madison, WI, USA). Moreover, skin inoculated with *S. aureus* and left untreated was used as a negative control, corresponding to the 0.00% iodine treatment. After treatment, the udder skin was placed in sterile Whirl-Pak® (Madison, WI, USA) bags and mixed with Buffered Peptone Water (BPW, Millipore Sigma, Burlington, MA, USA) at 230 rpm for 2 min. For microbial enumeration, 10-fold serial dilutions were performed and plated onto TSA plates in duplicate. Plates were incubated overnight at 37 °C, and colonies were counted and log-transformed for statistical analysis.

### 2.3. Iodine's Effect on S. aureus Biofilms

The eighteen isolates used in the previous test were also evaluated to determine the capacity of iodine, at different concentrations, to inhibit biofilm formation. An *S. aureus* strain (ATCC 43300) was incorporated into the study as a reference for biofilm formation. Each strain was treated with 0.25%, 0.38%, 0.50%, 0.75%, and 1.00% of the commercial teat dip previously used. The strains were streaked onto TSA (TSA, Remel, Lenexa, KS, USA) agar and incubated at 37 °C for 24 h. After incubation, one colony was inoculated on Brain Heart Infusion (BHI, Millipore Sigma, Burlington, MA, USA) broth and incubated at 37 °C for 48 h. The culture was diluted in 9.9 mL BHI broth, and 100 μL was added to 96-well plates in triplicate, along with 100 μL of iodine at 0.25%, 0.38%, 0.50%, 0.75%, and 1.00%. Then, plates were incubated at 37 °C for 48 h. Next, each well was washed twice with 300 μL of distilled water, and 200 μL of 90% ethanol was added to fix the biofilm for 15 min; ethanol was removed, and the plates were left to air dry for 1 h. A volume of 200 μL of 1% crystal violet was added to each well and allowed to stain the biofilm for 45 min, and the crystal violet was further removed by washing the wells with distilled water. The plates were left upside down overnight at room temperature to air dry and protected to prevent contamination. The following day, 200 μL of 90% ethanol was added to each well for 3 min to destain the crystal violet. Finally, 100 μL of the previous suspension was spotted into a new 96-well plate and placed in a microplate reader spectrophotometer (Synergy LX Multi-Mode, BioTek, Winooski, VT, USA) to measure its absorbance at 580 nm. The measure of absorbance was compared with control plates (untreated) and used to determine the effectiveness of lower iodine concentrations on biofilm formation. The experiment was performed across three different biological replicates.

### 2.4. Antibiotic Susceptibility Testing

Antimicrobial susceptibility was performed on all *S. aureus* isolates originally obtained ($n = 30$), corresponding to five isolates from each of the six mastitic samples. A Sensititre™ Gram Positive MIC Plate GPN3F was used (Sensititre™ ARIS HiQ™ System, Thermofisher, Waltham, MA, USA), which includes erythromycin, clindamycin, quinupristin/dalfopristin, daptomycin, vancomycin, tetracycline, ampicillin, gentamicin, levofloxacin, linezolid, ceftriaxone, streptomycin, penicillin, rifampin, gatifloxacin, ciprofloxacin, trimethoprim/sulfamethoxazole, and oxacillin + 2% NaCl. The isolates were tested as per the Clinical Laboratory Standard Center (CLSI) guidelines [24]. Table 1 provides a list of antibiotics, and their respective breakpoints, used in this study.

Isolates were streaked onto TSA agar and incubated at 37 °C for 24 h. This process was repeated twice for optimal recovery. Then, one colony was picked from the TSA plate, streaked onto 5% defibrinated sheep blood agar plates (BAPs), and incubated at 37 °C for 24 h.

Upon incubation, 1–5 colonies were picked with a sterile cotton swab and inoculated in 5 mL of sterile water with 0.85% NaCl. The bacterial suspension was standardized to $1$–$2 \times 10^8$ CFU/mL using a Sensititre™ nephelometer calibrated at 0.5 McFarland. Following this step, 10 μL of the suspension was added to Muller Hinton Broth (MH, Thermofisher, Waltham, MA, USA) and thoroughly mixed. The Sensititre AIM™ System (Thermo Fisher, Waltham, MA, USA) was used to automatically dispense 50 μL of the bacterial suspension in the 96-well GPN3F microtiter plates. The plates were incubated in a Sensititre™ ARIS HiQ™ System (Thermo Fisher, Waltham, MA, USA) at 35 °C for 18 h. For the improved detection of oxacillin-resistant *Staphylococcus* and vancomycin-resistant strains, the incubation period was extended to 24 h. Then, the system automatically read the MIC.

**Table 1.** Antibiotics listed in the GPN3F panel and their corresponding breakpoints according to the CLSI.

| Antibiotics | Breakpoints (µg/mL) | | |
|---|---|---|---|
| | Susceptible (S) | Intermediate (I) | Resistant (R) |
| Erythromycin (ERY) | ≤2 | 4 | ≥8 |
| Clindamycin (CLI) | ≤0.5 | 1–2 | ≥4 |
| Quinupristin/dalfopristin (SYN) | ≤1 | 2 | ≥4 |
| Daptomycin (DAP) | ≤1 | | |
| Vancomycin (VAN) | ≤2 | 4–8 | ≥16 |
| Tetracycline (TET) | ≤4 | 8 | ≥16 |
| Ampicillin (AMP) [1] | N/A | | |
| Gentamicin (GEN) | ≤4 | 8 | ≥16 |
| Levofloxacin (LEVO) | ≤1 | 2 | ≥4 |
| Linezolid (LZD) | ≤4 | | ≥8 |
| Ceftriaxone (AXO) [1] | N/A | | |
| Streptomycin (STR) [1] | N/A | | |
| Penicillin (PEN) | ≤0.12 | | ≥0.25 |
| Rifampin (RIF) | ≤1 | 2 | ≥4 |
| Gatifloxacin (GAT) | ≤0.5 | 1 | ≥2 |
| Ciprofloxacin (CIP) | ≤1 | | ≥4 |
| Trimethoprim/sulfamethoxazole (SXT) | ≤2/38 | | ≥4/76 |
| Oxacillin + 2%NaCl (OXA+) | ≤2 | | ≥4 |

[1] No breakpoint rule indicated in the CLSI for *Staphylococcus* spp. However, methicillin (oxacillin)-susceptible staphylococci can be considered susceptible to ceftriaxone. N/A: Non-Applicable

### 2.5. Testing S. aureus Isolates' Relatedness Using Microbial Typing

Microbial strain typing using a Fourier transform infrared (FTIR) spectroscopy system with the IR Biotyper® (IR-BT, Bruker corporation, Billerica, MA, USA), was used to determine the relatedness between strains. The purpose of this assay was to determine if the same isolate could be linked to all mastitis secretions as a potential sole source of infection. For consistency, the same 18 isolates (three per infected milk sample) as also used in previous experiments were subjected to typing. First, the strains were streaked onto TSA agar plates and incubated at 37 °C for $24 \pm 0.5$ h; the process was repeated twice. The samples were prepared for analysis following the manufacturer's instructions. Initially, a 1 µL colony-overloaded loop from the confluent part of the plate culture was resuspended in 50 µL of a 70% ethanol solution in an IR-BT vial. The suspension was vortexed, and 50 µL of deionized water was added before the solution was mixed by pipetting. Five technical replicates were spotted onto a 96-well silica plate, by inoculating 15 µL of the bacterial suspension on the corresponding spots, and left to dry for 10 min at 37 °C under environmental conditions. Infrared Test Standards (IRTSs) 1 and 2 of the IR-BT kit were resuspended in 100 µL of deionized water, and 60 µL of absolute ethanol was added and mixed. Then, 10 µL of the resulting suspension was spotted in duplicate onto the IR-BT target and left to dry as previously described. All steps for IR-BT sample preparation and measurements were performed at a microbiology laboratory bench without controlled room temperature and humidity conditions. Lastly, all samples underwent analysis alongside one independent biological replicate.

The spectra were obtained and processed using OPUS software V7.5.18 (Bruker Optics, GmbH, Germany) and IR Biotyper® Client Software V 4.0, using the default settings recommended by the manufacturer. The spectra were acquired in the wave number range of 800–1300 cm$^{-1}$, corresponding to the absorption region of polysaccharides. Principal Component Analysis (PCA) was employed to generate the variables' associations. In addition, quality control was carried out as per the manufacturer's instructions. Furthermore, a hierarchical cluster analysis (HCA) was conducted, and its similarity matrix and average linkage were calculated using the Euclidean and UPGMA method. Finally, the software

used the "SDI × mC" algorithm to determine a possible cutoff value for partitioning, which indicates the maximum distance up to which the spectra belong to the same group.

### 2.6. Statistical Analysis

Data analysis was performed using R statistical software version 4.3.0. (Lucent Technologies, Murray Hill, NJ, USA) This study used a completely randomized design with a factorial arrangement to assess the impact of commercial iodine treatments on the concentration of *S. aureus*. Two factors, namely iodine concentration and dipping time, were considered, which had six and four levels, respectively. The experimental unit consisted of the udder skin treated and all counts were transformed to Log CFU/g. The main effects of iodine concentration and dipping time and their interaction were evaluated using a two-way ANOVA. Post hoc analysis was accomplished using a pairwise *t*-test with a Bonferroni p-adjustment method for multiple comparisons. Moreover, a completely randomized design was used to assess the commercial iodine's effect on S. *aureus* biofilm formation. Significant differences in biofilm optical density at 580 nm among the various treatments were evaluated using the Kruskal–Wallis test. In instances where statistically significant results were obtained, the Wilcoxon rank sum test was employed as a substitute for the pairwise *t*-test. A significance level (alpha) of 0.05 was utilized to determine statistical significance.

### 2.7. Ethical Statement

This project did not involve the use of animals. Samples for this study were collected from culled cows (carcasses of passed cows) and obtained by the attending veterinarian for diagnostic purposes.

## 3. Results

A total of 30 isolates obtained from the six mastitis-infected cows' milk were targeted in this study to assess how specific conditions to control *S. aureus* could influence the dissemination of this bacterium. Each milk sample was obtained from different cows affected by the chronic mastitis outbreak, which led to their deaths. Five isolates per sample were recovered. Confirmation using a real-time PCR assay and MALDI-TOF indicated that all isolates were *S. aureus*.

### 3.1. Reduced Iodine Concentrations to Control S. aureus on Udder Skin

A standardized concentration of 1% iodine was reported by the dairy to be used for post-milking teat disinfection. Reduced iodine concentrations were tested to challenge the current treatment's effectiveness. This approach allowed us to identify the range of iodine concentrations at which the disinfectant maintained an efficacy against *S. aureus* comparable to the 1% iodine.

Dipping times showed no statistical significance ($p = 0.658$) across all the iodine concentrations employed in this study. Therefore, dipping time was removed as a main effect to provide a better description and visualization of the different iodine concentrations and their impact on the *S. aureus* counts in the samples. The results indicated that the fragments of udder skin not exposed to iodine (control samples) had an average bacterial attachment of 6.54 Log CFU/g (Table 2). The analysis also indicated a statistical difference ($p < 0.001$) between inoculated untreated udder skin and inoculated skin treated with different iodine concentrations. Following exposure to the treatments, iodine produced an average destruction of 3.06 Log CFU/g of *S. aureus* on the inoculated udder skin ($p < 0.001$), equivalent to a 99.9% reduction. As shown in Figure 1, differences ($p < 0.001$) were observed when comparing iodine concentrations of 0.25% and 0.38% to 1% iodine. However, no difference ($p > 0.137$) was found between samples treated with iodine concentrations of 0.5%, 0.75%, and 1%.

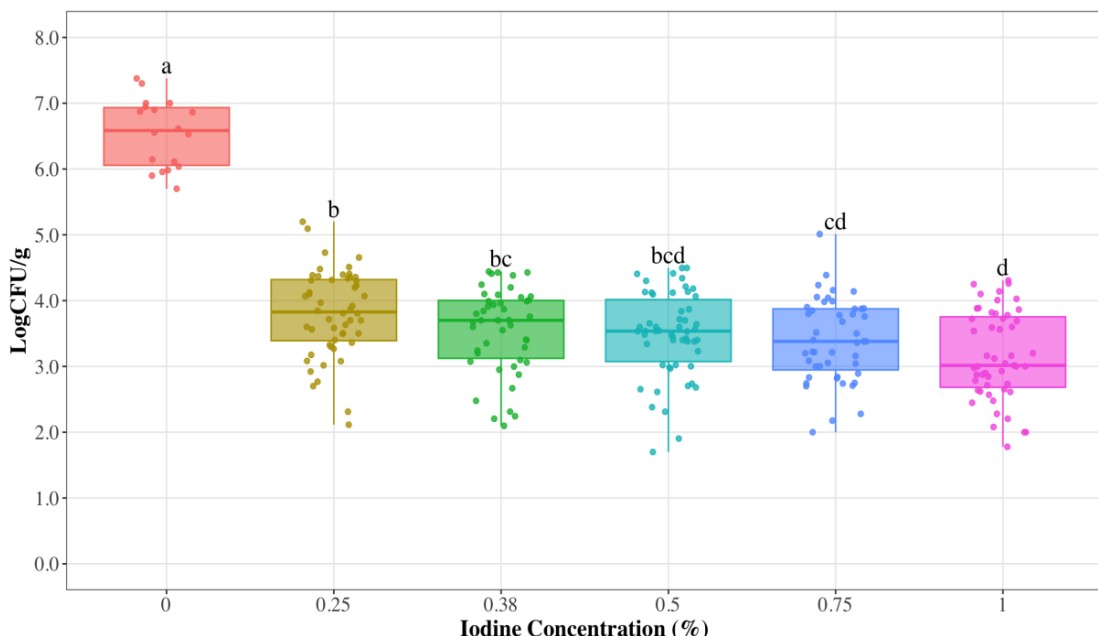

**Figure 1.** In vitro efficacy of commercial iodine in reducing *S. aureus* isolates on udder skin using different concentrations of 0.25% (yellow), 0.38% (green), 0.5% (blue), 0.75% (purple), and 1% (pink). The graph depicts individual data points as dots. The horizontal line within the boxplot represents the median. The upper and lower limits of the box represent the interquartile range, while the bars extending from the box represent values up to 1.5 times the interquartile range.

**Table 2.** *S. aureus* reductions on udder skin post-treatment across various iodine concentrations.

| Iodine Concentration (%) | N | *S. aureus* Concentration Log CFU/g $\pm$ SE [1] | Average Reduction Log CFU/g |
|---|---|---|---|
| 0.25 | 52 | 3.81 $\pm$ 0.09 | 2.73 |
| 0.38 | 46 | 3.55 $\pm$ 0.09 | 2.99 |
| 0.5 | 54 | 3.48 $\pm$ 0.09 | 3.06 |
| 0.75 | 47 | 3.39 $\pm$ 0.09 | 3.15 |
| 1 | 51 | 3.16 $\pm$ 0.09 | 3.38 |

[1] Standard error of the mean.

The response of *S. aureus* biofilms to iodine treatments was also determined. The average biofilm density of *S. aureus* in the absence of iodine was 0.59 $\pm$ 0.02, and this value was used as the point of reference for treatment responses (Table 3). The ANOVA revealed a significant effect of iodine concentrations on the biofilm density of *S. aureus* ($p < 0.001$), with an average biofilm formation of 0.09 $\pm$ 0.01 after treatment. As depicted in Figure 2, statistical difference ($p < 0.001$) was observed between the 0% iodine treatment compared to the different iodine concentrations applied. However, there was no statistical difference in the biofilm density of *S. aureus* between the 0.25%, 0.38%, 0.50%, 0.75%, and 1.00% iodine concentrations ($p = 0.230$). Overall, when iodine is not applied at any of the tested concentrations, the *S. aureus* isolates tend to produce at least 5.78 times more biofilm.

**Table 3.** Effect of iodine on *S. aureus*'s biofilm formation.

| Iodine Concentration (%) | N | Biofilm Concentration (OD $\pm$ SE [1]) | Biofilm Formation Odds Ratio |
|---|---|---|---|
| 0.25 | 18 | 0.09 $\pm$ 0.01 | 5.78 |
| 0.38 | 18 | 0.08 $\pm$ 0.00 | 6.50 |
| 0.50 | 18 | 0.08 $\pm$ 0.00 | 6.50 |
| 0.75 | 18 | 0.09 $\pm$ 0.01 | 5.78 |
| 1.00 | 18 | 0.09 $\pm$ 0.00 | 5.78 |

[1] Standard error of the mean.

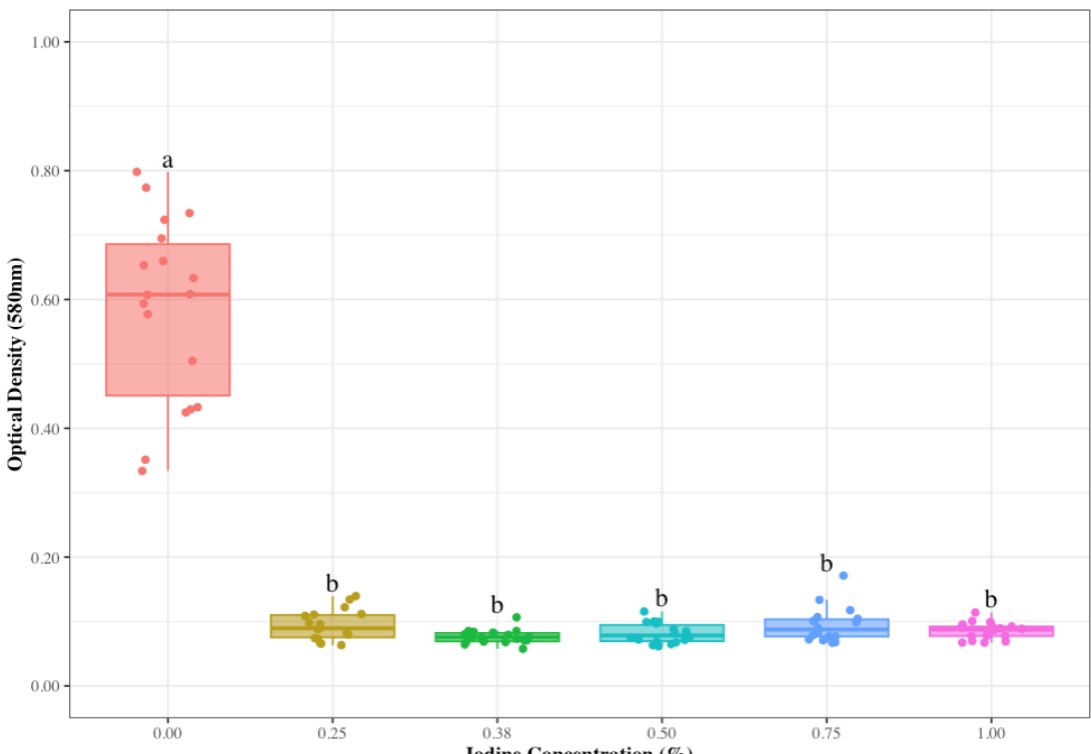

**Figure 2.** Biofilm density of *S. aureus* at different iodine concentrations (0.25% yellow, 0.38% green, 0.5% blue, 0.75% purple, and 1% pink. The graph depicts individual data points as dots. The horizontal line within the boxplot represents the median. The upper and lower limits of the box represent the interquartile range, while the bars extending from the box represent values up to 1.5 times the interquartile range. Boxplots with different letters indicate statistical differences ($p < 0.05$).

### 3.2. Antimicrobial Susceptibility Testing

A total of 30 isolates obtained from mastitis-infected milk, confirmed as *S. aureus*, were subjected to the broth microdilution method. Following their analysis, through broth microdilution and adhering to the CLSI guidelines, it was determined that none of the isolates exhibited resistance (Table 4).

**Table 4.** Antibiotic susceptibility distribution of *S. aureus* isolates using broth microdilution and the CLSI guidelines.

| Antibiotic | Exact (Clopper–Pearson) | | Minimum Inhibitory Concentration MIC (µg/mL) Distribution (*n* = 30) * | | | | | | | | | | | | | | | | | |
|---|---|---|---|---|---|---|---|---|---|---|---|---|---|---|---|---|---|---|---|---|
| | R (%) | 95% CI | 0.5/9.5 | 1/19 | 2/38 | 4/76 | 0.06 | 0.12 | 0.25 | 0.5 | 1 | 2 | 4 | 8 | 16 | 32 | 64 | 128 | 500 | 1000 |
| ERY | 0 | 0–0.1157033 | | | | | | | 30 | 0 | 0 | 0 | 0 | | | | | | | |
| CLI | 0 | 0–0.1157033 | | | | | | 30 | 0 | 0 | 0 | 0 | | | | | | | | |
| SYN | 0 | 0–0.1157033 | | | | | 1 | 29 | 0 | 0 | 0 | 0 | | | | | | | | |
| DAP | 0 | 0–0.1157033 | | | | | | | 30 | 0 | 0 | 0 | 0 | 0 | | | | | | |
| VAN | 0 | 0–0.1157033 | | | | | | | | 30 | 0 | 0 | 0 | 0 | 0 | 0 | 0 | 0 | | |
| TET | 0 | 0–0.1157033 | | | | | | | | | | 30 | 0 | 0 | 0 | | | | | |
| AMP | 0 | 0–0.1157033 | | | | | | 30 | 0 | 0 | 0 | 0 | 0 | 0 | | | | | | |
| GEN | 0 | 0–0.1157033 | | | | | | | | | | 30 | 0 | 0 | 0 | | | | 0 | |
| LEVO | 0 | 0–0.1157033 | | | | | | | 30 | 0 | 0 | 0 | 0 | 0 | | | | | | |
| LZD | 0 | 0–0.1157033 | | | | | | | 0 | 0 | 30 | 0 | 0 | 0 | | | | | | |
| AXO | 0 | 0–0.1157033 | | | | | | | | | | | | 30 | 0 | 0 | 0 | | | |
| STR (N/A) | | | | | | | | | | | | | | | | | | | | 0 |
| PEN | 0 | 0–0.1157033 | | | | | 30 | 0 | 0 | 0 | 0 | 0 | 0 | 0 | | | | | | |
| RIF | 0 | 0–0.1157033 | | | | | | | | | 30 | 0 | 0 | 0 | | | | | | |

**Table 4.** *Cont.*

| Antibiotic | Exact (Clopper–Pearson) | | Minimum Inhibitory Concentration MIC (µg/mL) Distribution (*n* = 30) * | | | | | | | | | | | | | | | | | |
|---|---|---|---|---|---|---|---|---|---|---|---|---|---|---|---|---|---|---|---|---|
| | R (%) | 95% CI | 0.5/9.5 | 1/19 | 2/38 | 4/76 | 0.06 | 0.12 | 0.25 | 0.5 | 1 | 2 | 4 | 8 | 16 | 32 | 64 | 128 | 500 | 1000 |
| GAT | 0 | 0–0.1157033 | | | | | | | | | 30 | 0 | 0 | 0 | | | | | | |
| CIP | 0 | 0–0.1157033 | | | | | | | | 30 | 0 | 0 | | | | | | | | |
| STX | 0 | 0–0.1157033 | 30 | 0 | 0 | 0 | | | | | | | | | | | | | | |
| OXA+ | 0 | 0–0.1157033 | | | | | | | 30 | 0 | 0 | 0 | 0 | 0 | | | | | | |

* Grey fields indicate antibiotic concentrations not tested. White fields represent antibiotic concentrations tested and susceptible isolates. R: resistance, ERY: erythromycin, CLI: clindamycin, SYN: quinupristin/dalfopristin, DAP: daptomycin, VAN: vancomycin, TET: tetracycline, AMP: ampicillin, GEN: gentamicin, LEVO: levofloxacin, LZD: linezolid, AXO: ceftriaxone, STR: streptomycin, PEN: penicillin, RIF: rifampin, GAT: gatifloxacin, CIP: ciprofloxacin, SXT: trimethoprim/sulfamethoxazole, OXA+: oxacillin + 2%NaCl.

### 3.3. S. aureus Strains' Relatedness

Three isolates per milk sample were used for fingerprint analysis and to test the relatedness between strains. Figure 3 presents the dendrogram representing the spectra results, which depict the samples identified as cow 1 through 6 (indicating each milk sample from the deceased cows), and isolates were labeled from 1 to 18 (identified with the letter "I"). Isolates were analyzed using IR-BT spectroscopy to construct a dendrogram with the average spectra. Two bacterial strains, *S. aureus* (UKB 07) and *Streptococcus* spp. (UKH_0886), were incorporated as reference isolates for comparative analysis. The dendrogram showed that 13 out of the 18 isolates showed significant phenotypic similarities and were classified into a single cluster. Despite the high degree of coherence, cophenetic similarity, and correlation in the dendrogram (0.993), isolates I1, I3, I4, I5, and I6 did not fall within this cluster, which were one isolate from cow 3 (I9) and two from cows 1 (I1 and I3) and 2 (I4–I6). The exclusion of these isolates from the main cluster was based on the cut-off value of 0.075, determined by the software used the Euclidean and UPGMA method. However, it is noteworthy that at least one isolate from each cow was still categorized within the same group, indicating a possible common source of infection. Furthermore, both reference isolates were markedly different from those in the study.

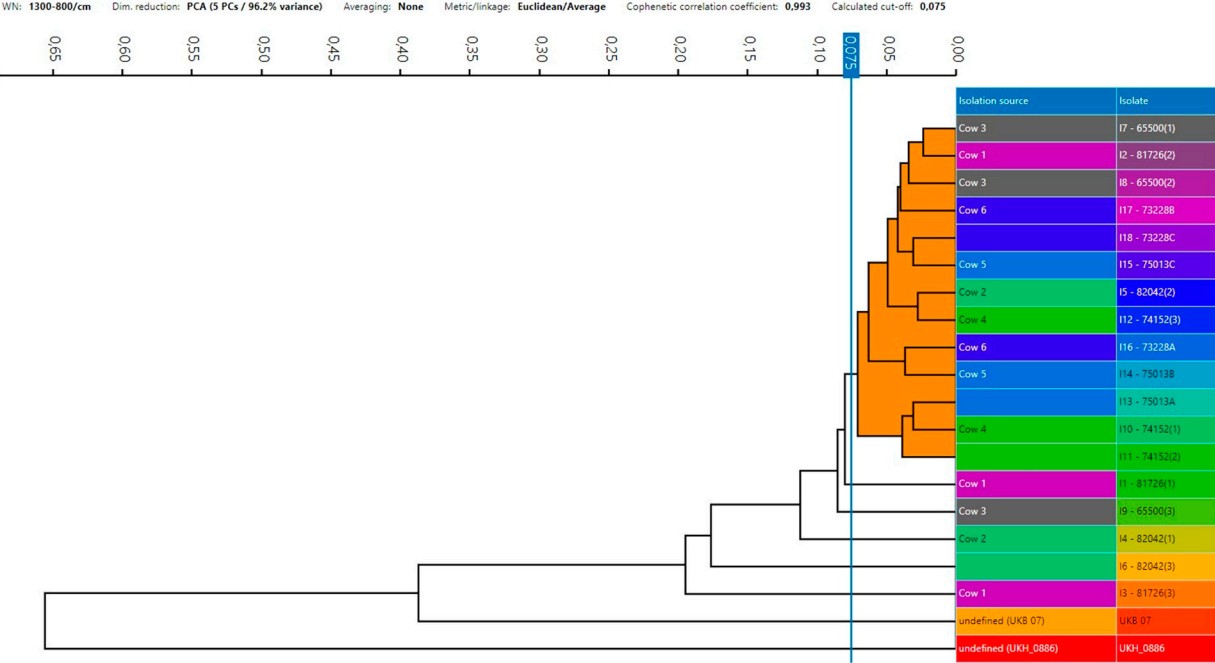

**Figure 3.** Dendrogram of 20 average spectra from PCA analyses. Eighteen *S. aureus* isolates obtained from cattle with chronic mastitis infection and two reference isolates (UKB 07, and UKH_0886).

## 4. Discussion

This study explored the impact of varying concentrations of iodine teat dips on *S. aureus* found in udder secretions from cows affected by a severe outbreak of deadly mastitis infections. Reduced concentrations of iodine as a teat disinfectant were tested.

Using iodine teat dips is a common and well-established practice in dairy operations. Previous research has highlighted that the response to teat dips can vary among different strains within bacterial species, making it an important area of investigation [25,26]. In this study, the iodine treatments were effective ($p < 0.001$) at reducing all *S. aureus* strains tested on the udder skin compared to the control, regardless of the concentration and dipping time. Furthermore, reducing the concentration to 50% of that currently used by the dairy in this study achieved a 99.9% microbial reduction. This exhibited comparable efficacy to the highest iodine concentration (1%). Consistent with these results, a study conducted in Poland demonstrated that the use of iodine was effective (<90%) at reducing Gram-positive bacteria, with a specific reduction of 99.3% of *S. aureus* on teat skin [27]. Other studies have also demonstrated that applying iodine at concentrations as low as 0.5% can effectively inactivate pathogens such as *S. aureus* [28,29]. Iodine's ability to kill bacteria is due to its capacity to oxidize different parts of bacterial cells, such as amino and fatty acids, nucleotides, lipids in the cell membrane, and enzymes in the cytosol. This oxidation leads to the denaturation and deactivation of these cellular components [30]. Therefore, considering iodine's mode of action, even lower iodine concentrations may hold promise in preventing *S. aureus* infections.

Moreover, the growth of staphylococci biofilms in infected tissues has been considered as another reason for the failure of therapy treatment, as various studies have demonstrated that this growth leads to innate resistance to most different therapeutic agents [31]. *S. aureus* biofilm formation involves a complex interplay of adhesion mechanisms, extracellular polymeric substance (EPS) matrix production, and regulated developmental stages, contributing to the bacterium's pathogenicity and antibiotic resistance [32,33]. Other studies have demonstrated that antimicrobials have little to no effect on the biofilm of bacteria challenging the effectiveness of treatments in mastitis infections in cows [34–36]. However, this study indicated that concentrations of 0.25% and 0.38% of iodine significantly reduced the biofilm formation in the *S. aureus* strains tested. This effect has also been observed in another study, where it was reported that 0.4% and 1% iodine effectively reduced the biofilm of staphylococcal isolates [37]. These data suggest that iodine might affect the biofilm formation of pathogens on the udder skin. Instead, other factors, such as the correct application of the disinfectant before and after the process, need to be considered as they could influence the effectiveness of teat disinfection, potentially preventing the mastitis infections caused by *S. aureus*.

The frequent use of antibiotics in the farm environment is believed to contribute to the rise of antibiotic-resistant organisms. Antibiotics are crucial in dry cow therapy, a common method used to prevent mastitis infections. Antimicrobial resistance in *S. aureus* is a significant global issue in managing mastitis infections. However, antibiotics' limited efficacy during lactation and difficulty penetrating the udder when *S. aureus* is established are factors to consider [38–41]. Surprisingly, all *S. aureus* isolates tested were susceptible to the antibiotics evaluated in this investigation. The findings from our study align with those of a previous investigation conducted on dairy farms in Tennessee, which observed that *S. aureus* strains isolated from mastitis-infected animals across different farms were also susceptible to the tested antibiotics. Despite the frequent identification of antimicrobial-resistant strains in dairy cattle with subclinical mastitis, reports have also indicated the coexistence of susceptible *S. aureus* strains in infected cattle [42,43]. Beyond antimicrobial resistance, the persistent mastitis associated with *S. aureus* is further linked to additional virulence factors that contribute to its pathogenicity and establishment within the mammary gland. These factors include its capacity to adapt and elude the host immune system, engagement in biofilm formation, and production of toxins, facilitating bacterial attachment and colonization [44–46].

Furthermore, the IR-BT technique was employed to assess the *S. aureus* isolates' relatedness for a more comprehensive understanding of the interrelation among them and their potential role in the infected cattle. This technique resulted in the clustering of most bacterial strains, highlighting a substantial similarity among the isolates recovered from the diseased cattle population. It is worth noting that previous research has established a correlation between disease incidence within a herd and the prevalence of specific strain genotypes, often leading to the dominance of one or a few distinct clones. This phenomenon suggests the potential for transmission between animals and that there is a preference for particular pathotypic traits [47]. However, the cluster did not include five of the isolates. It is important to note that other studies have established *S. aureus*'s relatedness at cut-off values of 0.215 [48], whereas this study used a cut-off value of 0.075. This variation in cut-off values might explain the exclusion of some isolates from the group.

These *S. aureus* isolates were collected during a major outbreak of severe mastitis among the cattle on the farm. Despite the application of antimicrobial treatments, the outbreak proved challenging to control, raising concerns about the potential resistance of *S. aureus* to antimicrobials. However, this investigation showed that these isolates were susceptible to the tested antimicrobials. This observation has prompted the hypothesis that other virulence factors beyond antimicrobial resistance might be influencing the persistence of *S. aureus* in the mammary glands. It has been documented that *S. aureus* does not only evade host immune defenses but that it also internalizes within host cells, leading to its persistence, intracellularly, for long periods of time [49,50].

Animal-to-animal transmission is crucial in *S. aureus*-induced mastitis within dairy farming. Key factors such as close contact with infected quarters, shared equipment, contaminated bedding, improper milking, and inadequate hygiene can create opportunities for the exchange of infectious agents [4]. Therefore, it is necessary to underscore the significance of implementing enhanced hygiene practices during the milking process, improving personnel cleaning procedures, ensuring the segregation of cows with mastitis in dedicated hospital pens, and carefully evaluating the possibility of culling cows with chronic infections to reduce morbidity and mortality in the herd. [51–53]. These measures can be crucial in disease control and preventing further outbreaks within the farm environment.

## 5. Conclusions

These findings indicate that iodine remains effective even at lower concentrations and that other reasons for microbial spread contributed to the dissemination of the organism in the environment. The *S. aureus* isolates responsible for the infections during this mastitis outbreak did not represent a challenge when treated with iodine or antibiotics. While appropriate antibiotic stewardship is necessary during food animal production, this research suggests that other factors might contribute to *S. aureus*'s persistence in mastitis infections. The presence of at least one isolate in the same cluster implies that identical resident strains within the farm can disseminate throughout the herd and contribute to mastitis outbreaks. The *S. aureus* isolates in this study were biofilm formers, and this attribute facilitates the organisms' formation of niches and harbors within the environment. Considering that the iodine disinfectant solution and antibiotics were effective, farms are recommended to focus on an essential deep disinfection of their facilities.

**Author Contributions:** Conceptualization, A.C.; Methodology, A.P. and R.J.; Formal analysis, M.S.; Investigation, A.P., M.S. and R.J.; Resources, A.C.; Data curation, A.P.; Writing—original draft, A.P. and M.S.; Writing—review & editing, A.C.; Supervision, A.C.; Project administration, A.C. All authors have read and agreed to the published version of the manuscript.

**Funding:** This study received no external funding.

**Data Availability Statement:** Data are contained within the article.

**Acknowledgments:** The authors would like to thank Diego Casas and Babafela Awosile for their guidance with the statistical analysis.

**Conflicts of Interest:** The authors declare no conflicts of interest.

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
