# Peer review of "The Tale of Staphylococcus aureus Isolated from Mastitis Infections: The Effect of Antimicrobials and Bacterial Relatedness"

_2673-8007, doi:10.3390/applmicrobiol4010035_

Round 1

Reviewer 1 Report

Comments and Suggestions for Authors

The sentence "For this portion of the study, eighteen S. aureus isolates were selected, representing three isolates from each of the six original mastitic milk samples" could be clearer. So, was the study actually conducted on only 3 out of 6 milk samples? Would it be a study in triplicate instead of a study of 18 samples?

In line 170, it mentions that 170 S. aureus isolates were obtained. Were these isolates from 6 milk samples? The sampling methodology is not clear.

Again, I am in doubt about the sampling, in section 2.5. Testing S. aureus Isolates Relatedness using Microbial Typing, why were 3x6 (total of 18 samples) used instead of 30? Is it possible to make a statistical inference of correlation from this sampling?

It is undeniable that the study used excellent diagnostic techniques. However, for a proper evaluation of the article's reproducibility and the applicability of the results obtained, a better understanding of the sampling is necessary.

Author Response

Response to Reviewer #1 comments and recommendations:

  1. The sentence "For this portion of the study, eighteen S. aureus isolates were selected, representing three isolates from each of the six original mastitic milk samples" could be clearer. So, was the study actually conducted on only 3 out of 6 milk samples? Would it be a study in triplicate instead of a study of 18 samples?

Author response: We understand that more clarity will be necessary. We have improved the sentence to make it more clear to the reader. However, we would like to explain to the reviewer that we received six samples (from six deceased cows), and from each, we isolated multiple S. aureus strains. For this portion of the study, we selected only three isolates obtained per each of the six mastitic milk samples.

The paragraph was improved, and it currently reads:

“For this portion of the study, three isolates were selected from each of the six original mastitic milk samples (n=18).”

  1. In line 170, it mentions that 170 S. aureus isolates were obtained. Were these isolates from 6 milk samples? The sampling methodology is not clear. 

Author response: In line 170, we indicated that we used 30 isolates for the antibiotic susceptibility test. In this case, we used all isolates recovered from the six samples with the purpose of increasing the sample size.

We would like to mention that in lines 103 and 104, we have already indicated that five isolates were recovered from each of the six samples, for a total of 30 (presumptive) S. aureus recovered during this project. The present research had a total of 30 isolates. In some cases, only 18 isolates (3 per each of the six samples) were used, and in other cases, 30 isolates (5 per each of the six samples) were used. In each portion of the study, we have indicated whether we tested 30 or 18 isolates. The decision to use 18 (3x6) vs 30 (5x6) was based on the specific experiment. In general, we maintained consistency in all experiments using the same 18 isolates; only for the AMR, we tested all 30 isolates.

We made the following updates to the document to improve clarity:

  • We have added the following text to the methods, lines 124-126:

“Thus, from the six original infected milk samples, a total of 30 isolates (5 isolates per sample) were recovered. From these 30 isolates, a subset of 18 (3 isolates per each of the six samples) were used in some of the experiments, as described further.”

  • Lines 128 and 129 were updated:

“For this portion of the study, three isolates were selected from each of the six original mastitic milk samples (n=18).”

  • Lines 179-180 were modified:

“Antimicrobial susceptibility was performed on all S. aureus isolates obtained originally (n=30), corresponding to five isolates per each of the six mastitic samples.”

  1. Again, I am in doubt about the sampling, in section 2.5. Testing S. aureus Isolates Relatedness using Microbial Typing, why were 3x6 (total of 18 samples) used instead of 30? Is it possible to make a statistical inference of correlation from this sampling?

Author response: We hope that with the previous clarifications about the methods, the amount of S. aureus isolates used is better understood. We are willing to continue making changes if needed.  We believe the previous modifications contribute to a better understanding of methods in section 2.5. We have made an update in lines 214-215:

“For consistency, the same 18 isolates (three per infected milk sample), also used in previous experiments, were subjected to typing.”

  1. It is undeniable that the study used excellent diagnostic techniques. However, for a proper evaluation of the article's reproducibility and the applicability of the results obtained, a better understanding of the sampling is necessary.

Author response: We understand how selecting 30 or 18 isolated could be confusing. Sampling was carried out only one time during a mastitis outbreak, which was explained in lines 96-97.

We expect the new text added in lines at the end of the materials and method portion to help clarify the isolates selection. If the sampling requires more revisions, we are willing to make more changes.

Lines 131-133: “Thus, from the six original infected milk samples, a total of 30 isolates (5 isolates per sample) were recovered. From these 30 isolates, a subset of 18 (3 isolates per each of the six samples) were used in some of the experiments, as described further.”

Reviewer 2 Report

Comments and Suggestions for Authors

Dear editor and authors,

thank you for the opportunity to review interesting manuscript:  The tale of Staphylococcus aureus isolated from mastitis infections: Effect of antimicrobials and bacterial relatedness. The manuscript has potential for application and meets the requirements for publication in Applied Microbiology. The authors made several mistakes that I have highlighted in the following comments:

The title of the article is suitable.

Abstract

It is not clear from the abstract how many cows were examined, how many were positive and how many S. aureus isolates were tested.

Introduction

In the introduction, I recommend adding a text describing disinfectant distribution during the udder toilet (pre and post dipping).

Lines 36, 39, 60: After the sentences, I recommend adding more recent citations: https://www.mdpi.com/2076-2615/12/4/470

https://www.mdpi.com/2306-7381/10/6/386

Lines 69 – 82: The part in the introduction describing S. aureus is useless because it does not bring anything new.

Materials and methods

The part with the description of the farm where the practical part of the study was carried out is completely missing. Number of milked cows and hygienic milking program. How were samples taken from deceased cows?

Lines 103 – 105: The procedure for further identification of S. aureus is not clear. Why did the authors take 5 colonies from each plate when it was from one sample (one deceased cow). It is okay that the authors considered the sample positive, but it is still one strain. S. aureus strains cannot be genotypically different within one sample.

Results

Figure 1,2 and the table 2 do not show a comparison of the tested isolates with the reference strain of S. aureus  (ATCC 43300)

Antimicrobial Susceptibility Testing is poorly presented without expression breakpoints (μg/mL) in tested strains.

Discussion

In the discussions, elaborate more on the formation of S. aureus biofilm. I also recommend comparing the results for biofilm formation with several studies.

Conclusion

The conclusion is clear and reflects the most important findings.

Author Response

Response to comments and recommendations - Reviewer 2

  1. Abstract - It is not clear from the abstract how many cows were examined, how many were positive, and how many S. aureus isolates were tested.

Author response: We have added a few extra words in the abstract to improve clarity.

“During this investigation, a total of 30 S. aureus isolates were obtained from six deceased cows, all diagnosed with chronic mastitis from an outbreak in West Texas.”

  1. Introduction - In the introduction, I recommend adding a text describing disinfectant distribution during the udder toilet (pre and post dipping).

Author response: Considering your valuable recommendation, we have added a text to add a description of the use of iodine.  This text can be found in lines 53-56:

“Teat disinfection usually occurs by conducting a teat dip using iodine at 1%. Upon milking, udders are always sealed and protected with this antimicrobial solution to prevent microbial attachment and multiplication.”

  1. Lines 36, 39, 60: After the sentences, I recommend adding more recent citations: https://www.mdpi.com/2076-2615/12/4/470

https://www.mdpi.com/2306-7381/10/6/386

Author response: We appreciate the recommendation. They are great papers and recent information. We have added two citations as we found they fit our manuscript perfectly.

  1. Lines 69 – 82: The part in the introduction describing aureus is useless because it does not bring anything new.

Author response: We appreciate the comment and recommendation. After reviewing and consideration, the authors decided to move the paragraph to the beginning of the introduction to provide context to less expert readers. We agree that this paragraph, as well as other information presented in the introduction, does not provide new information; however, it provides background and context pertaining to S. aureus and mastitis infections.

  1. Materials and methods - The part with the description of the farm where the practical part of the study was carried out is completely missing. Number of milked cows and hygienic milking program. How were samples taken from deceased cows?

Author response: we have added more information to describe the sample collection and current outbreak experienced by the dairy farm.  To improve clarity and provide more information, we have added a text that can be found in lines 98-105:

“The attending veterinarian manually extracted the milk secretions from infected quarters from the already deceased cows, deposited them in sterile cups, and delivered to the Food Microbiology Laboratory at Texas Tech University (TTU). The dairy farm providing samples was a medium to large operation, located in west Texas, experiencing a mastitis outbreak. Infected cows were segregated in the hospital pen for treatment or culling.  Animal segregation served as a means to prevent spread of the bacteria infection among the herds.”

  1. Lines 103 – 105: The procedure for further identification of S. aureus is not clear. Why did the authors take 5 colonies from each plate when it was from one sample (one deceased cow). It is okay that the authors considered the sample positive, but it is still one strain. S. aureus strains cannot be genotypically different within one sample.

Author response: We appreciate the observation. In discussions with veterinarians, we have learned that multiple strains and coexisting organisms may infect animals at the same time, causing mastitis infections. In addition, from past experiences with other research projects, we have collected more than one colony from a given sample and found more than one strain of the same bacterial species (data not published) present on the same agar plate. Considering that, we subjected the samples to microbial typing –as one of our objectives was to detect if several S. aureus strains were responsible for the infections. Our results indicate that bacterial isolates were grouped within more than one cluster.  

We hope this clarifies the point, and we are open to providing more information pertaining to our rationale.

  1. Results - Figure 1,2 and the table 2 do not show a comparison of the tested isolates with the reference strain of S. aureus (ATCC 43300)

Author response: We appreciate your review and the opportunity to address your concerns. Regarding using the reference strain S. aureus (ATCC 43300), the strain was not incorporated into the statistical analysis to compare it with our results because, as explain the methods, the ATCC strain was used as a reference to determine if our tested strains were biofilm formers. After identifying that they all produced biofilm, we proceeded with the laboratory and statistical analysis, evaluating the inhibition of biofilm and comparing between concentrations. 

  1. Antimicrobial Susceptibility Testing is poorly presented without expression breakpoints (μg/mL) in tested strains.

Author response: Based on your comment, we have carefully reviewed the analysis and agree that an improvement will be beneficial in this portion. We have designed and incorporated Table 4 to represent the obtained results in a more detailed and comprehensive manner. This new table describes the minimum inhibitory concentration per each antibiotic obtained from the tested strains. We present the isolates that were susceptible to each antibiotic.

In addition, during the corrections made to the manuscript, we identified a mistake in how a portion of the results were presented.  An additional update was made, presented in lines 337-339.  The paragraph reads:  “A total of 30 isolates obtained from mastitis-infected milk confirmed as S. aureus were subjected to the broth microdilution method.  Following analysis through broth microdilution and adhering to the CLSI guidelines, it was determined that none of the isolates exhibited resistance (Table 4)”

  1. Discussion - In the discussions, elaborate more on the formation of S. aureus biofilm. I also recommend comparing the results for biofilm formation with several studies.

Author Response: Following careful consideration, the authors have incorporated the suggested text, in lines 424-428:  "S. aureus biofilm formation involves a complex interplay of adhesion mechanisms, extracellular polymeric substance (EPS) matrix production, and regulated developmental stages, contributing to the bacterium's pathogenicity and antibiotic resistance."

Additionally, a text describing how results pertaining to biofilm formation can compared with several other studies is presented in lines 434 – 439. That paragraph highlights the consistency with findings from other relevant studies.

  1. Conclusion - The conclusion is clear and reflects the most important findings.

Round 2

Reviewer 2 Report

Comments and Suggestions for Authors Thanks for incorporating all my comments into the text. However, it is not clear to me how the authors demonstrated in the results that antimicrobial resistance does not always play a role in persistent mastitis infections caused by S. aureus as is described in the abstract. The results and discussions should further elaborate on this part of the study.    

Author Response

Dear Reviewer,

Thank you for the observation. After reviewing, we realized that more clarity, as you indicated, was needed. We agree that antimicrobial resistance may play a role in persistent mastitis; however, our results did not show significant data that supports that fact. Persisten strains within a dairy farm and persistent mastitis, is also affected by other factors associated with colonization and establishment of the organisms in the environment and on the tissues. Since we did not find major AMR among the isolates, we conducted literature review and found other papers with similar results. 

We made some updates that can be found between lines 399-415:

"The frequent use of antibiotics in the farm environment is believed to contribute to the rise of antibiotic-resistant organisms. Antibiotics are crucial in dry cow therapy, a common method to prevent mastitis infections. Antimicrobial resistance in S. aureus is a significant global issue in managing mastitis infections However, their limited efficacy during lactation and difficulty penetrating the udder when S. aureus is established are factors to consider [38-41]. Surprisingly, all S. aureus isolates tested were susceptible to the antibiotics evaluated in this investigation. The findings from our study align with a previous investigation conducted on dairy farms in Tennessee, which observed that S. aureus strains isolated from mastitis-infected animals across different farms were also susceptible to the tested antibiotics. Despite the frequent identification of antimicrobial-resistant strains in dairy cattle with subclinical mastitis, reports have also indicated the coexistence of susceptible S. aureus strains in infected cattle [42, 43]. Beyond antimicrobial resistance, the persistent mastitis associated with S. aureus is further linked to additional virulence factors that contribute to its pathogenicity and establishment within the mammary gland. These factors include its capacity to adapt and elude the host immune system, engagement in biofilm formation, and production of toxins, facilitating bacterial attachment and colonization [44-46]."